**Data Availability Statement:** All relevant data are contained within the paper and/or supporting information files.

**Funding:** R01 DK120844 NIDDK (RAK) The funders had no role in study design, data collection

# Social and maternal behavior in mesoderm specific transcript (*Mest*)-deficient mice

Rea P. Anunciado-Koza[1☯], J. Patrizia Stohn[1☯], Arturo Hernandez[1,2,3], Robert A. Koza[1,3,4]*

1 Center for Molecular Medicine, Maine Medical Center Research Institute, Scarborough, Maine, United States of America, 2 Department of Medicine, Tufts University School of Medicine, Boston, MA, United States of America, 3 Graduate School of Biomedical Sciences and Engineering, University of Maine, Orono, ME, United States of America, 4 Pennington Biomedical Research Center, Baton Rouge, LA, United States of America

☯ These authors contributed equally to this work.
* Robert.Koza@Mainehealth.org

## Abstract

Mesoderm specific transcript (*Mest*)/paternally expressed gene-1 (*Peg1*) is an imprinted gene expressed predominantly from the paternal allele. Aberrations in maternal behavior were previously reported in a *Mest* global knockout mouse (*Mest*tm1Masu). In this study, we performed in-depth social and maternal behavioral testing in a mouse model of *Mest* inactivation developed in our laboratory (*Mest*tm1.2Rkz). Mice with paternal allele inactivation (*Mest*pKO) did not show anxiety after testing in the elevated plus maze, open field trial, and marble burying; nor depression-like behaviors in the tail suspension test. *Mest*pKO showed normal social behaviors and memory/cognition in the three-chamber box test and the novel object recognition test, respectively. Primiparous *Mest*pKO and *Mest*gKO (biallelic *Mest* inactivation) female mice exhibited normal nest building and maternal behavior; and, virgin *Mest*pKO and *Mest*gKO female mice showed normal maternal instinct. Analyses of gene expression in adult hypothalamus, embryonic day 14.5 whole brain and adult whole brain demonstrated full abrogation of *Mest* mRNA in *Mest*pKO and *Mest*gKO mice with no effect on miR-335 expression. Our data indicates no discernible impairments in object recognition memory, social behavior or maternal behavior resulting from loss of *Mest*. The basis for the differences in maternal phenotypic behaviors between *Mest*tm1Masu and *Mest*tm1.2Rkz is not known.

## Introduction

Mouse mesoderm specific transcript (*Mest*), also known as paternally expressed gene 1 (*Peg1*), is an imprinted gene that maps to mouse *Chr 6* and is expressed in a strict parent-of-origin-specific differential methylation pattern [1, 2]. The inactive maternal allele of *Mest* is methylated in a CpG-rich region containing exon 1 and is not expressed in the embryo, whereas the active paternal allele is unmethylated and fully expressed [2]. Sequence analysis of MEST

and analysis, decision to publish, or preparation of the manuscript

**Competing interests:** The authors have declared that no competing interests exist.

suggests it encodes a protein with homology to the alpha/beta hydrolase fold family signifying an enzymatic role [1, 3].

Although the function of MEST is unknown, its role in adipose tissue has been well characterized [4–6]. *Mest* mRNA expression has been shown to vary up to ~80-fold in white adipose tissue (WAT) from individual C57BL/6J mice fed an obesogenic diet and variations of adipose *Mest* mRNA (and protein) showed a positive association with the rate of fat mass deposition [5–7]. Inactivation of the paternal allele of *Mest* (*Mest*$^{pKO}$) in mice either globally or in adipocytes resulted in reduced dietary fat induced adipose tissue expansion and adipocyte hypertrophy, improved glucose tolerance, and reduced WAT expression of genes associated with hypoxia and inflammation compared to high fat fed wild type (WT) littermate controls [4]. Detailed *in vitro* studies in our laboratory using immunofluorescence confocal microscopy showed that MEST is located in the endoplasmic reticulum (ER) membranes with considerable co-localization to the lipid droplet surface protein perilipin at distinct ER-lipid droplet contact points during adipogenesis [8].

*Mest* has also been shown to be highly expressed in the brain of mice during both development [9] and as adults [10], suggesting that it could play a role in the regulation of behavior [9].

A previously developed global *Mest* knock-out mouse (*Mest*$^{tm1Masu}$) model was reported to show reduced postnatal survival rates; and, female *Mest*-deficient mice exhibited impaired maternal behavior [10]. In these earlier studies, *Mest* mutant females showed decreased reproductive fitness and substantial perinatal loss of mutant pups at birth. The poor survival of mutant pups was due to failure of the *Mest*-deficient dams to respond to the newborns together with impaired placentophagia. In contrast, *Mest*$^{tm1.2Rkz}$ mice generated by our laboratory showed the expected Mendelian frequencies in litters produced by heterozygous female and male breeding regimens [4]. In this study, our goal was to evaluate social and maternal behaviors of the *Mest*$^{tm1.2Rkz}$ mutant mice.

## Materials and methods

### Mouse breeding and study design

We used *Mest*$^{tm1.2Rkz}$ mice with a global inactivation generated from animals with a floxed *Mest* allele (*Mest*$^{tm1.1Rkz}$) for our studies. The strategy for gene targeting and the validation of *Mest* allele inactivation have been described in detail [4]. These mice are fully congenic on a C57BL/6J genetic background. *Mest*$^{tm1.1Rkz}$ is now available from The Jackson Laboratory (Strain #036527). Since *Mest* is a maternally imprinted, paternally expressed gene, heterozygous mice that carry the inactivated *Mest* on the maternal allele (KO/+; *Mest*$^{mKO}$) express fully intact *Mest* mRNA and protein, whereas mice that carry the inactivated *Mest* on the paternal allele (+/KO; *Mest*$^{pKO}$) have complete absence of MEST and are comparable to homozygous knockout (KO/KO; *Mest*$^{gKO}$) mice. *Mest*$^{pKO}$ mice were generated by crossing WT female with *Mest*$^{mKO}$ or *Mest*$^{pKO}$ males yielding 50% WT and 50% *Mest*$^{pKO}$ offspring (Fig 1A). Mice with biallelic deletion of *Mest* (*Mest*$^{gKO}$) were generated by crossing heterozygous *Mest* knockout mice (Fig 1B). We observed that *Mest*$^{gKO}$ mice breed well and when intra-crossed were able to generate a homozygous KO colony. Fourteen litters (153 *Mest*$^{gKO}$ pups; 10.9 pups per litter) were produced and yielded the expected gender ratio (76 female/77 male) (Chi-square = 0.007, P = 0.9356).

All testing procedures used in this study were reviewed and approved by the Maine Medical Center Research Institute Institutional Animal Care and Use Committee in accordance with National Institutes of Health guidelines for care and use of laboratory animals. Mice were kept in a barrier facility under standard light conditions and fed 2018 Teklad Global 18% Protein

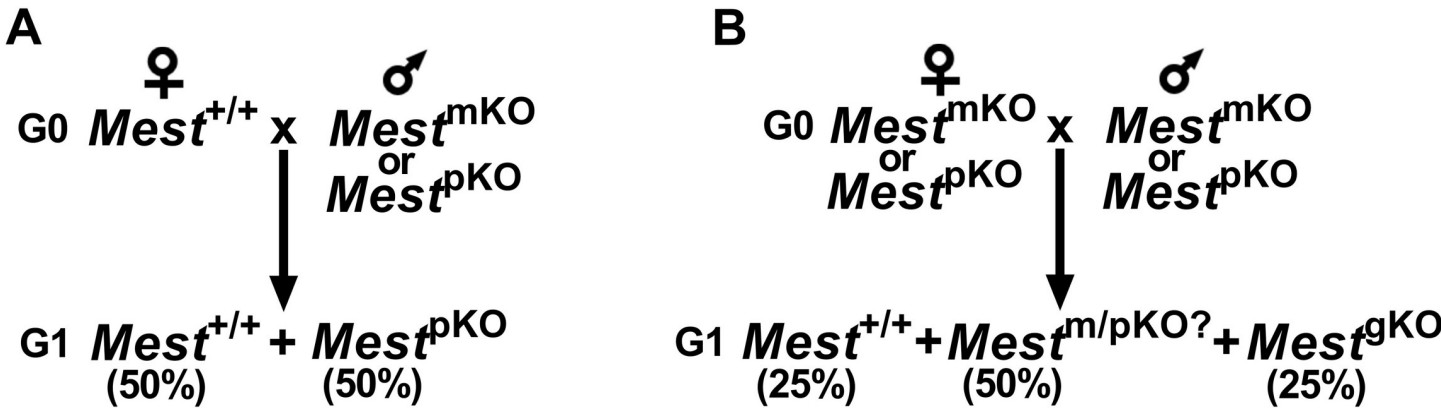

**Fig 1. Breeding strategy.** Breeding schemes to generate $Mest^{pKO}$ (a) and $Mest^{gKO}$ (b) mice.

Rodent Diet (Envigo, Indianapolis, IN, USA) post-weaning. Mouse room temperature was maintained at 23–24˚C.

## Behavioral testing

Behavioral tests were conducted in WT and $Mest^{pKO}$ mouse littermates of both sexes between 12–14 weeks of age with n = 13–15 per group. Mice were group-housed (2–4 animals per cage) after weaning. The temperature of the behavioral test room was kept consistent with the animal room at 23–24˚C. The experimental animals were placed in the testing room for two to three hours prior to each behavioral test and left undisturbed during this period to acclimate. Mice underwent a battery of tests for anxiety, depression, object recognition memory and social behaviors. Tests were conducted in the following order: elevated plus maze (day 1), marble burying (day 5), tail suspension test (females: day 8; males: day 9), open field test (females: day 12; males: day 14), novel object recognition for 2 days (females: days 13–14; males: days 15–16), and social behaviors (females: days 18, 22 and 24; day 20 in males). In addition, virgin female mice were subjected to nestlet shredding/nest building and maternal behavior tests 2 weeks after culmination of social behavioral testing. The virgin females were mated 1 week after completion of tests.

The test apparatuses were purchased from Stoelting (Wood Dale, IL, USA) and experiments were recorded either with the ANY-maze™ video tracking system v5.14 (Stoelting, Wood Dale, IL, USA) or a camcorder (Samsung HMX-F90, Seoul, South Korea). Videos were analyzed offline with the ANY-maze™ software or The Observer XT v12.5 from Noldus (Leesburg, VA, USA).

## Elevated plus maze

This test was performed using the elevated plus maze (EPM), which has a cross shape. Opposing arms were either enclosed by walls or open. The mouse was placed facing the center area of the cross to start the test and recorded for 5 min, while exploring the maze freely. The animal's movement was tracked. Time spent in the open, speed, and immobility were analyzed. Mice that fell off the maze were excluded from the tests and subsequent analysis.

## Marble burying test

A cage with the dimensions 45 cm by 23 cm was used and filled with a 5–6 cm deep layer of corncob bedding (Envigo, Indianapolis, IN, USA). Twenty-eight black marbles were spaced

out evenly on top of the bedding in five alternating rows of six and five. A mouse was placed in the cage and allowed to explore freely for 30 min under low lighting. The mouse was then removed gently from the cage at the end of the test to avoid disturbing the bedding. Marbles buried by at least two thirds of their diameter or more were counted as buried.

### Tail suspension test

Mice were suspended by their tail and fixed with adhesive tape to a shelf, so their heads would be located approximately 8 cm above a flat surface. Animals caged together were tested in parallel. Mice were video recorded for 5 min. An investigator blind to the genotype of the mice analyzed the recordings and scored the latency to become immobile, the episodes of immobility, the duration of these episodes and the total time that the mice spent immobile during the test.

### Open field test

We used a square setup with opaque walls for the open field test (OFT). The center area was defined as a 20 x 20 cm$^2$, while the remaining area was referred to as periphery. At the beginning of the test, the animal was placed in the periphery facing the center and allowed to explore the maze freely for 5 min. The animal's movement was tracked and time spent in the center of the open field was analyzed.

### Novel object recognition

The novel object recognition test (NOR) was a three-day test using the same apparatus as in the open field test. Therefore the OFT also served as habituation time and the NOR followed immediately upon it. On the first day of the NOR, two of the same objects (either rubber pineapple or ceramic pumpkin) were placed in two corners of the center area (see OFT) diagonally from each other. To start the test, a mouse was placed in one of the corners of the square apparatus and 2 different objects were placed in corners equidistant from the mouse on the left and right-hand side. The animal and object placements were alternated for each mouse to prevent location favoritism. The animal was tracked while moving around freely in the apparatus. On the second day, one of the objects was replaced with a different one, so that there was one known and one unknown object in the apparatus. Otherwise, the test was performed exactly as on day one. The initial object as well as the locations in the apparatus were set-up in alternates. The time mice spent examining the objects was analyzed and comparisons were made between mice and the different days.

### Sociability and social novelty (three chamber social box)

Sociability and social novelty were assessed using a three-chambered box similar to that previously described [11]. The test apparatus was a rectangular box (60 cm x 40 cm x 22 cm) made of clear plastic material. The outside walls were covered in a white opaque adhesive foil to reduce any potential outside distractions. Two walls divide the apparatus into three equally sized compartments that are connected by centered openings. The test comprised of three consecutive trials of 10 minutes each. At the beginning of each test, the mouse was placed in the center box of the apparatus and allowed to explore it freely, and in between trials, the mouse was returned to its home cage. The first trial is the habituation phase and evaluates any potential biases that might interfere with the actual testing. The second trial investigates sociability by evaluating the test subject's interest in other mice compared to an inert novel object. Therefore, two wired cylinders were placed in the left and right chamber. One of the cylinders

contained an unfamiliar mouse (stranger) of the same sex and age like the test mouse, while the second remained empty. The cylinder permitted the animals to sniff each other, but otherwise restricted physical contact. To test for social novelty in the third trial, a second stranger was placed in the empty cylinder, while the first, now familiar stranger, remained in its position. The position for the first stranger alternated between tests. Movement of the test mouse was tracked during all trials. Time spent in the different chambers was calculated and compared.

## Nest building

Nest building behavior was tested in both virgin and pregnant females. Pregnant females close to birth (2–3 days prior) or virgin females were moved a clean cage. A nestlet (Ancare Corporation, Bellmore, NY, USA) was weighed and placed in one of the corners at the back of the cage. The animal was returned to its shelf. After 24 hours, the nestlet was examined and any unused (not shredded) part was weighed. If the mouse used any of the provided material to build a nest, the structure was scored based on a 0 to 4 point system described by Deacon (2006) [12] with 0 = no nest and 4 = high walls, a nearly perfectly built nest.

## Maternal behavior

Maternal behavior testing was performed after all of the social behavior tests. Virgin females were tested initially. Post-testing, the virgin females were set-up for timed pregnancy. Pregnant females were then individually housed at 14 days of gestation and monitored for pre-partum and post-partum behaviors. After parturition, primiparous dams were tested for maternal behavior.

Virgin female mice were tested 2 days after the nestlet building test. The virgin female was removed from its home cage and temporarily placed in a holding cage. Primiparous dams were tested 2 days after parturition with her litter removed from the home cage and placed in a separate cage. Five of her pups were used in the testing. For the virgin females, 2–4 day old pups from a WT breeding pair was used. Pups were placed in the home cage in the pattern of six on a die with the nest being one of the spots. The primiparous dam or virgin female was then placed back in their nest or nest-like structure at the beginning of the test and their behavior was recorded for five minutes. The test was scored off-line for time sniffing the pups, pup retrieval and latency, total retrieval time, and digging or burying behavior around, or of the pups.

## Gene expression analyses

Hypothalami from 8-week old WT and $Mest^{pKO}$ mice were collected for analysis of gene expression. The following breeders were set-up to generate embryos for harvest: WT x WT, WT females x $Mest^{gKO}$ males, $Mest^{pKO}$ females x $Mest^{gKO}$ males and $Mest^{gKO}$ females x $Mest^{gKO}$ males. The morning a vaginal plug was noticed was considered embryonic day 0.5. Whole brains from embryos were harvested at day 14.5 of gestation (E14.5). A piece of embryonic tissue was obtained at dissection to confirm genotype and sex (*Sry*, sex determining region of Chromosome Y). Harvested tissues for RNA extraction were collected and snap-frozen in liquid nitrogen. Samples were stored at -70°C until processed and QRT-PCR was performed as described previously [4]. Briefly, RNA was isolated using TriReagent (Molecular Research Center, IN, USA) followed by purification using RNeasy Mini Kit and RNAse-free DNAse (Qiagen, Hilden, Germany). Extracted RNA was protected against RNAse contamination using SUPERase-In (Life Technologies, MA, USA). One-step qRT-PCR amplification was performed with TaqMan RNA to cT 1-Step reagent (Life Technologies) using a BioRad

CFX384 Real Time system (BioRad, Hercules, CA, USA). RNA quality and quantity was determined using Nanodrop 2000 spectrophotometer (ThermoFisher, Waltham, MA, USA). *Mest* mRNA expression was normalized to TATA binding protein (*Tbp*). Primer and probe sequences used to measure *Tbp* were: forward primer, 5′– `CTTCGTGCAAGAAATGCTGAAT-3′`; reverse primer, 5–`CAGTTGTCCGTGGCTCTCTTATT-3′` and probe, 5′–`FAM-TCCCAAGCGATT TGCTGCAGTCATC-BHQ-3′`, and for *Mest*; forward primer, 5′–`AAGCCATGTAAAAGCAC AACTATCTC-3′`; reverse primer, 5′–`CCTACAAAGGCCTACGCATCTT-3′` and probe, 5′–`FAM-TTCCGACCACACCGACAGAATCTTGG-BHQ-3′`.

QPCR for miRNAs was performed as previously described [4]. Total RNA including miR-NAs was isolated from brain using miRNeasy mini kit (Qiagen) and protected against RNAse contamination using SUPERase-In (Life Technologies). cDNA synthesis was performed for each sample, and included negative RT control and non-template control using the TaqMan MicroRNA Reverse Transcription Kit and TaqMan miRNA specific primers (Life Technologies) for miR-335-5p (Assay ID 000546) and snoRNA202 (NCBI Accession #AF357327; Assay ID 001232). cDNA was amplified using the TaqMan Universal Mastermix II, no UNG and TaqMan Small RNA Assay (Life Technologies). Total RNA was quantified as described earlier. QPCR was performed using CFX384 Real Time PCR detection system (BioRad) platform. Relative abundance of miR-335 was quantified using the 2-ΔΔCT method using snoRNA202 for normalization of expression.

## Western blot analysis

Total tissue lysate was isolated from gonadal fat as previously described [4]. Gonadal fat was pulverized under liquid nitrogen and homogenized in RIPA buffer containing protease inhibitors (Millipore Sigma, Darmstadt, Germany). For the brain samples, membrane fraction was enriched following a modified procedure from Harris et al [13]. Brain pulverized under liquid nitrogen was incubated in 200 ul of 60% sucrose dissolved in lysis buffer (pH 7.4: 10 mM HEPES, 1 mM) on ice for 10 minutes. Lysis buffer (800 ul) was added and incubated on ice for 10 minutes; and, homogenized using Teflon pestle. Another 600 ul of lysis buffer was added and homogenate was transferred to a 2 ml microfuge tube; and, centrifuged at 3000*g* at 4˚C for 5 minutes to remove large particulates. The supernatant was collected and transferred into another 2 ml microfuge tube; and centrifuged at 20,000*g* at 4˚C for 2 hours to obtain the membrane fraction, which was solubilized in RIPA buffer with protease inhibitors. Protein content was quantified using BCA assay (ThermoFisher, Waltham, MA, USA). Western blotting was performed as described previously [8] with the Mini-Protean Tetra Cell and Mini Trans-Blot Electrophoretic Transfer cell (Bio-Rad, Hercules, CA, USA). Nitrocellulose blots (GE Healthcare, Chicago, IL, USA) were incubated with antibodies against MEST (ABCAM ab151564, Boston, MA, USA; rabbit monoclonal, 1:2500) and calreticulin (Cell Signaling Technology #12238, Danvers, MA, USA; rabbit monoclonal, 1:10,000). Membranes were blocked for 5–6 hours at room temperature in 2% fish gelatin (Millipore Sigma, Darmstadt, Germany) in PBS-T (0.1%). Membranes were incubated overnight at 4˚C in the primary antibody and then secondary antibody at RT for 1 hour. Antibodies were diluted in Signal Boost Immunoreaction Enhancer (Millipore Sigma). Signals were detected by chemiluminescence (WesternSure Premium Chemiluminescent Substrate, Li-Cor, Lincoln, NE, USA). Blots were exposed to X-ray Film.

## Statistics

All data were analyzed using GraphPad Prism v. 9.3.1. Data from the EPM, OFT, marble burying, NOR, sociability and social novelty tests were analyzed using a 2-way ANOVA followed

by Sidak's test. Data from the nest building and maternal tests in virgins or dams was analyzed using either a one-way ANOVA followed by Tukey's test or a Chi-square test.

## Results

### Anxiety and despair

In order to assess the social and maternal behaviors in our *Mest* mutant mouse strain (*Mest*^tm1.2Rkz^), we performed a series of tests for anxiety and depression-like behaviors, object recognition memory, sociability, and social novelty. Furthermore, we evaluated nestlet shredding and nest building as well as maternal behavior in virgins and primiparous dams.

It is well recognized that increased anxiety and stress can lead to altered maternal behavior in mammals [14]. Therefore, to test whether these behavioral modalities may contribute to maternal behavior in our *Mest*^pKO^ mice, we subjected these mice to various tests to evaluate potential differences in anxiety levels in this mutant strain. The animals underwent testing in the elevated plus maze, the open field test and the marble burying test. Main indicators for increased anxiety are the avoidance of the open arms of the elevated maze or the bright center area of the open field test as well as an increased digging behavior, which is caused by the mouse's need to escape or hide. We did not find any significant differences for these parameters, suggesting that there are no abnormalities in the levels of anxiety in either female or male *Mest*^pKO^ mice compared to WT littermate controls (Fig 2A–2C).

Another possible effect on maternal behavior could be caused by altered depression-like or despair behavior. The tail suspension test is a typical test to measure this in mice. In this test mice are hanging by their tails above a surface. Typically the mice will actively try to escape this situation whereas 'giving up' the struggle and stopping movement is an indicator for

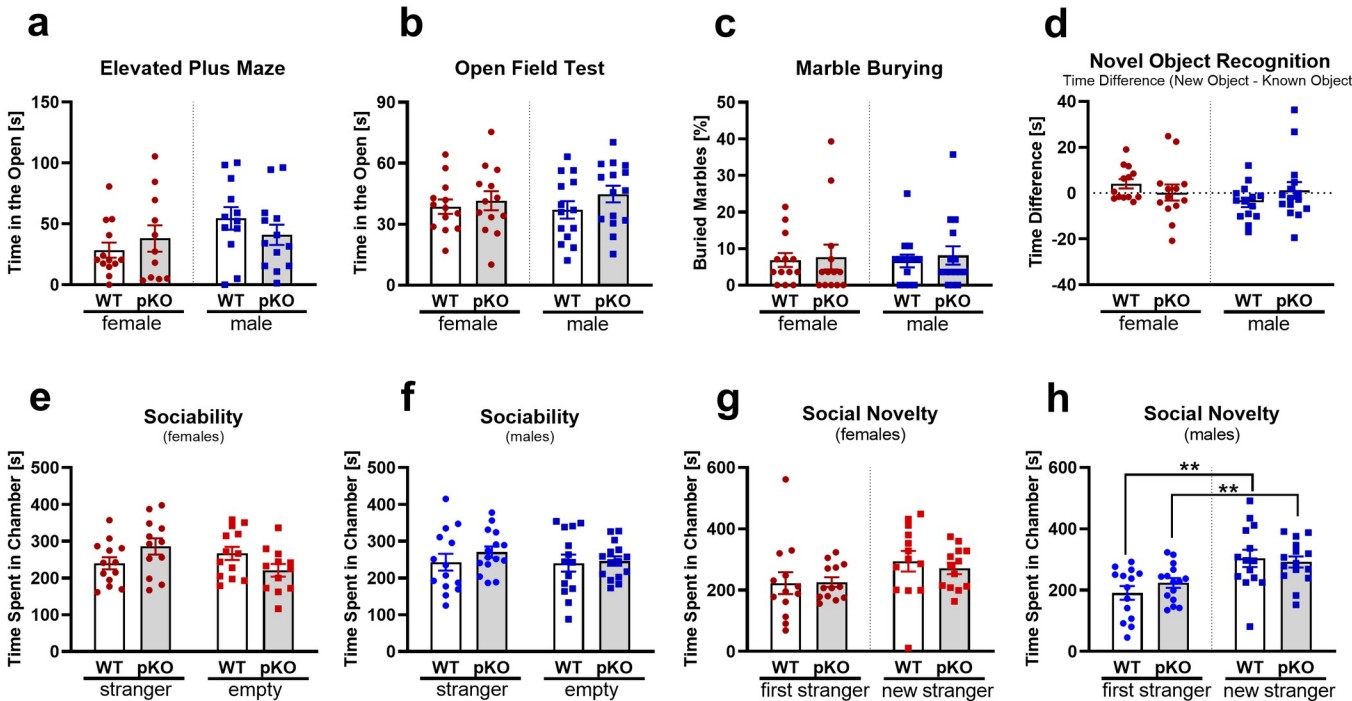

**Fig 2. Social behavioral testing.** *Mest*^pKO^ mice showed no abnormalities in elevated plus maze (a), open field test (b), marble burying (c), novel object recognition (d), sociability (e, f) and social novelty (g, h) tests. 11–15 mice were analyzed per group and significance between groups was determined using an unpaired t-test. Data with 2 asterisks indicate P ≤ 0.01.

depression-like behavior. When undergoing the tail suspension test, the majority of the animals (~85%) climbed up on their tail, which is a common phenomenon in some strains of mice and is not indicative of depression-like behavior [15]. Only two of the females and six of the male mice did not execute this behavior. The latency and time of immobility in those eight animals was very similar between sex and genotypes (S1 Table), suggesting that $Mest^{pKO}$ mice do not exhibit altered despair behaviors.

## Object recognition memory

We examined object recognition memory in mice using the NOR test. During the first day of this test, mice were introduced to two identical objects placed diagonally from each other and allowed to investigate them. On the second day, one of the objects was replaced with a new object to determine whether the mice would recognize the familiar object and show increased interest in the new object. Our results showed no differences in time investigating the familiar or new object for either sex or genotype (Fig 2D). However, there is some concern that during the 300s long test on the first day, the mice spent only 8-20s investigating one of the objects, which may not have long enough to memorize it. Although the animals had been habituated to the empty maze a day prior to the NOR test, it is possible that the new surroundings of the maze were too much of a distraction for the mice. While a longer test time on day 1 may have allowed mice to better memorize the objects, there was no difference in behavior between WT and $Mest^{pKO}$ mice using our method which suggests that NOR is invariant between genotypes.

## Social behavior

An important aspect to consider are social behaviors. Initially, the three chamber social test was used to examine the reaction of a mouse with an unknown mouse, which is normally stationary in a mini cage within one of the chambers. This allows for interaction between mice but avoids physical conflicts or aggression. These studies showed that $Mest^{pKO}$ mice tend to spend more time in the chamber with the cage containing the unknown mouse than the chamber with an empty cage compared to the WT mice (Fig 2E and 2F). While there were no significant differences between genotypes, these data do suggest that the sociability of $Mest^{pKO}$ mice is at least similar to WT mice regardless of sex. The second part of the analyses focuses on social novelty by placing another stranger in the empty cage. In all our groups, the test subjects showed increased interest in the new mouse over the already familiar one, demonstrating normal interest in social novelty (Fig 2G and 2H).

## Virgin females: Maternal behavior

Maternal behavior in $Mest^{tm1Masu}$ knockout mice has been previously described as abnormal [10]. We tested maternal behavior in dams and virgins of different genotypes (WT, $Mest^{pKO}$ and $Mest^{gKO}$) and observed nest building skills in our model. First, we investigated maternal behavior in virgin females. Since these mice have never been mated, these tests are designed to evaluate the maternal instincts in these animals. Two days prior to exposing pups to females, the females were moved to a clean cage and provided with nesting material. The mice did not utilize the nesting material to build tall nests; however, they appeared to use it for padding their sleeping and resting spot. On the day of the test, the mouse was removed from the home cage and three 2-day old pups from a WT breeding set-up were placed in all the corners that were distant from the nest-like structure. The female was returned to the cage onto the padded area that she built and observed for maternal behavior. All virgin females showed interest in the pups (100% sniffing the pups) and some of the females began to retrieve them to their "nest". While not statistically significant ($X^2_{(4)} = 5.676$, P = 0.225), both $Mest^{pKO}$ (64.7%) and

$Mest^{gKO}$ (43.8%) virgin females showed increased frequency of any type of pup retrieval (partial or complete) compared to WT (31.6%) virgin females (Fig 3A). In addition, $Mest^{pKO}$ virgin females also trended to perform better with completing the retrieval all the pups compared to WT and $Mest^{gKO}$ virgin females (Fig 3A). The latency of pup retrieval (Fig 3B) among mice that did retrieve pups within the 300 second (5 min) period of time for this study was not significantly different between genotypes ($F_{(2,21)} = 1.831$; p = 0.185). We observed digging behavior around the pups and/or burying the pups in the bedding. This type of threat and aggressive behavior was likely triggered by the stress of the intrusion and the exposure to the unfamiliar pups, and is more pronounced in the virgins compared to the dams (see below). However, there was a significant decrease ($X^2_{(2)} = 6.734$, P = 0.035) of more than 50% in digging around and burying pups in the $Mest^{pKO}$ (35.3%) and $Mest^{gKO}$ (37.5%) virgin mice compared to WT (73.7%) virgin mice (Fig 3C).

## Primiparous dams: Maternal behavior

Next, we mated the virgin females. Although the conception rate trended to be lower in the $Mest^{gKO}$ female mice (56.3%) compared to WT (84.2%) or $Mest^{pKO}$ (82.4%) females (Fig 4A), Chi-square analyses across groups showed no significant differences($X^2_{(2)} = 4.350$, P = 0.114). A few days prior to the expected parturition, the pregnant females were moved to a clean cage and provided with nesting material. Upon inspection 24 hours later, the females had constructed similar padded areas as recently observed in the virgin mice. Two days after giving birth, we conducted the maternal behavior test. The dam was briefly removed from its home cage and the nest and litter size were scored prior to removing some of the pups from the cage. Five of the pups were left scattered in the corners and the middle of the home cage outside the nest while the remaining pups maintained in a separate cage during the test. The dam was then returned to the nest of the home cage and her behavior observed. Nests of high quality are an important factor for the survival of the newborn pups. While the virgin and pregnant females failed to build tall structures, the late-term/post-partum females created highly rated tall nests that covered and protected their offspring. No significant differences ($F_{(2,33)} = 2.785$; p = 0.0763) in the nest building skills were observed among the genotypes (Fig 4B). Furthermore, all dams tested

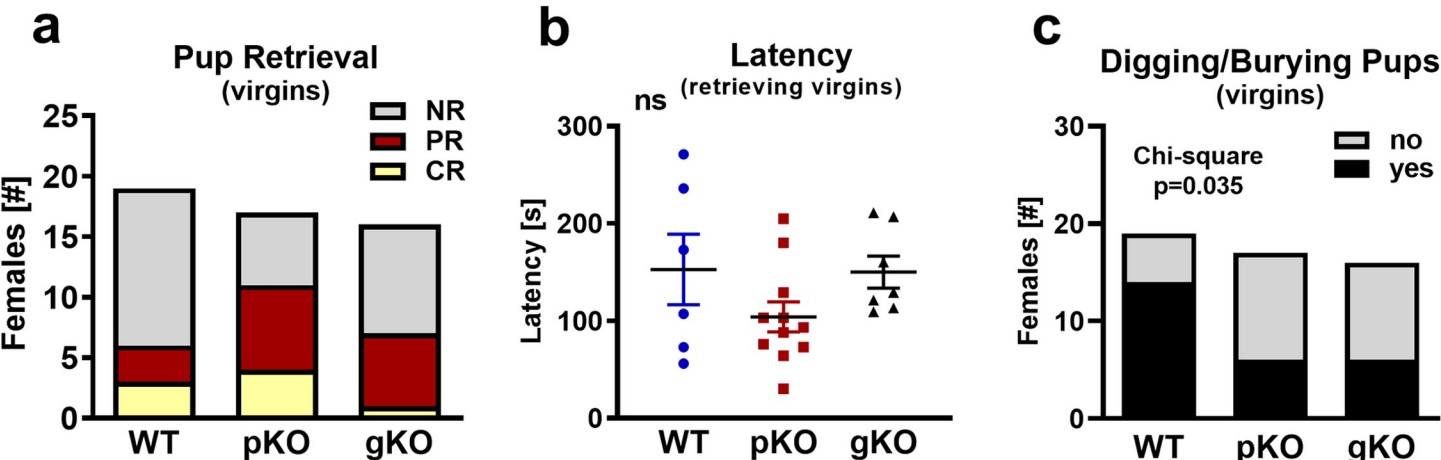

**Fig 3. Maternal behavior testing in virgin female mice.** $Mest^{pKO}$ (n = 17) and $Mest^{gKO}$ (n = 16) virgin females showed improved pup retrieval (a, b) and reduced digging/burying of pups (c) compared to wild-type (n = 19) mice. Analyses of latency, time to retrieve the first pup, (b) utilized 6-WT, 11-$Mest^{pKO}$ and 7-$Mest^{gKO}$ virgin females that retrieved pups within 5 min. Data were analyzed either by Chi-square (a, c) or multiple one-way ANOVA (b). NR, no retrieval; PR, partial retrieval; CR, complete retrieval.

retrieved their pups to the nest with no observed differences in latency (Fig 4C; $F_{(2,33)}$ = 1.402; p = 0.261) or total retrieval time (Fig 4D; $F_{(2,31)}$ = 2.037; p = 0.148). However, we noted that on average, it took the WT dams the longest to retrieve all of their pups and one of the WT dams did not finish the retrieval during the duration of the test. We also observed digging and burying behavior in some of the WT and $Mest^{pKO}$ females (Fig 4E). These kind of threat and aggressive behaviors in the mothers could be due to stress and anxiety caused by the preceding separation from their pups. Although not significant ($X^2_{(2)}$ = 3.718, P = 0.156), this stress-induced aggression was reduced by more than half in the $Mest^{pKO}$ mice when compared to the WT (14.3% vs. 31.3%, n.s.), and this finding is similar to the decrease we already observed in the virgins. This behavior was not observed in the $Mest^{gKO}$ dams.

Overall, the maternal instinct in the $Mest^{pKO}$ and $Mest^{gKO}$ virgins appears to be comparable to WT females. While levels of anxiety caused by the introduction to unknown pups are generally higher in the virgins compared to the primiparous dams (73.7% vs. 31.3%), they are significantly lower in the $Mest^{pKO}$ and $Mest^{gKO}$ females compared to the WT. In addition, $Mest^{pKO}$ virgins showed a higher rate of pup retrieval as well as reduced latency to bring pups into their "nest".

### *Mest* and miR-335 in hypothalamus and whole brain

To determine whether *Mest* gene expression matched with the genotypes of the mice, we analyzed adult hypothalamus and embryonic whole brain. *Mest* gene expression analysis in adult

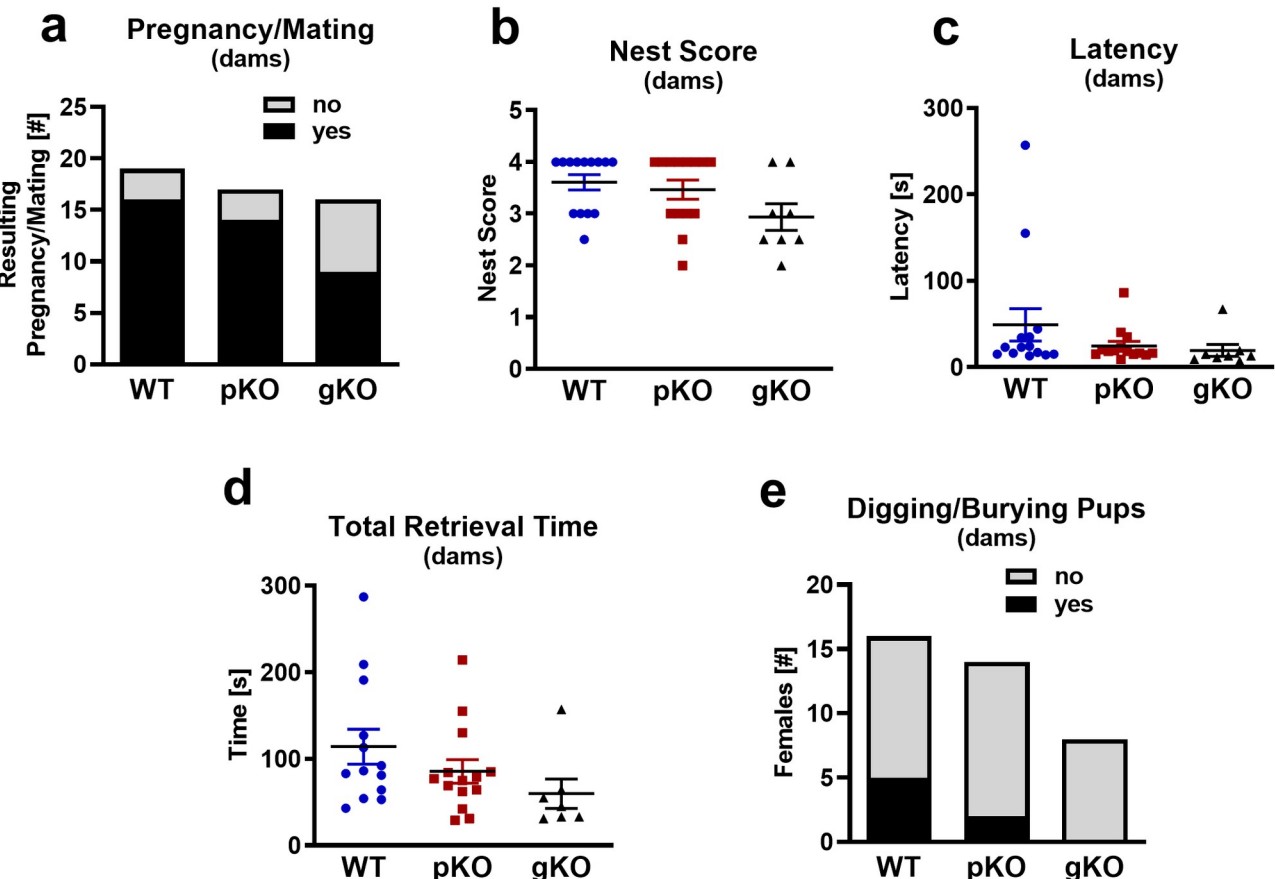

**Fig 4. Maternal behavior testing in primiparous female mice.** Conception rate (a), nest building (b), latency, time to retrieve the first pup (c), total pup retrieval time (d) and pup digging/burying behavior (e) did not differ in wild-type, $Mest^{pKO}$ and $Mest^{gKO}$ primiparous mice. Data were analyzed either by Chi-square (a, e) or multiple one-way ANOVA (b, c, d). No significant differences were observed between groups.

hypothalamus showed 394-fold higher gene expression in WT mice compared to $Mest^{pKO}$ mice (Fig 5A). In depth gene expression analysis in whole brain collected at E14.5 showed significantly reduced $Mest$ mRNA expression in $Mest^{pKO}$ (reduced >1600-fold) and $Mest^{gKO}$ (reduced >20,000-fold) compared to WT embryos from both sexes (Fig 5B). Although $Mest$ mRNA expression was low in $Mest^{pKO}$ mice compared to WT, it was >10 fold higher compared to mice with a global biallelic inactivation of $Mest$ ($Mest^{gKO}$) suggesting a very small (~0.25%) but measureable contribution of $Mest$ mRNA transcribed from the maternal allele. Furthermore, analyses of MEST protein using endoplasmic reticulum (ER)-enriched membrane fractions isolated from adult brain shows an absence of MEST in $Mest^{pKO}$ mice (Fig 5C). Additionally miR-335, a microRNA located in an intron between exons 2 and 3 of the $Mest$ allele, showed no differences of expression in brain of WT vs $Mest^{pKO}$ or $Mest^{gKO}$ mice (Fig 5D) and no association with $Mest$ mRNA (Fig 5E). Overall, our data demonstrates that $Mest$ is almost exclusively expressed from the paternal allele in the fetal and adult brain and that the brains of $Mest^{pKO}$ mice are essentially devoid of MEST. However, global or paternal inactivation of $Mest$ in our mouse model ($Mest^{tm1.2Rkz}$) shows no aberrant effect on the expression of miR-335.

## Discussion

Proper maternal behavior is crucial for survival of the newborn offspring. Maternal behavior includes those exhibited by the dam in preparation for the arrival of the newborn, in the care and protection of the newly arrived young, and the weaning of their young [14]. Although the first time mother is immediately and appropriately maternal, a virgin with no prior exposure to young does not necessarily show the immediate and appropriate behavior to foster young [14].

MEST has been proposed to function in the regulation of mammalian behavior based on its expression in neuronal tissue [9]. $Mest$ is abundantly expressed in the adult brain of mice with high levels in the hypothalamus, amygdala, ventral hippocampus and olfactory bulbs [10]. MEST and its intron product, miR-355 have been reported to play a critical role in brain development by affecting neuronal migration in the neocortex [16]. MEST has also been shown to be involved in the development and maintenance of a subset of neurons in the substantia nigra of the midbrain [9]. The exact function of MEST, especially in the brain, remains to be determined.

A previously developed mouse model with a global inactivation of $Mest$ ($Mest^{tm1Masu}$) showed reduced postnatal survival rates; and, female $Mest$-deficient mice exhibited impaired maternal behavior [10]. In these earlier studies, $Mest$ mutant females showed decreased reproductive fitness with perinatal loss of one-third of mutant pups at birth. The poor survival of mutant pups was due to failure of the $Mest$-deficient dams to respond to the newborns together with impaired placentophagia.

The current studies herein indicate that our mouse model of $Mest$ inactivation ($Mest^{tm1.2Rkz}$) exhibited no abnormalities in anxiety and depression-like behaviors, object recognition memory, sociability, and social novelty behaviors compared to WT mice. Batteries of tests are routinely used to investigate complex and multivariate neurological diseases/behaviors that cannot simply be described by a single test. However, although we selected an order for testing to reduce the potential that they do not affect the outcome of each other, there is always a possibility that performing a series of behavioral tests, in addition to repeated handling of mice within a 20–24 day period may not be ideal [17]. However, regardless of these potential limitations, except for possible differences in digging and burying pups (Fig 3C), no significant behavioral differences were observed between mouse strains.

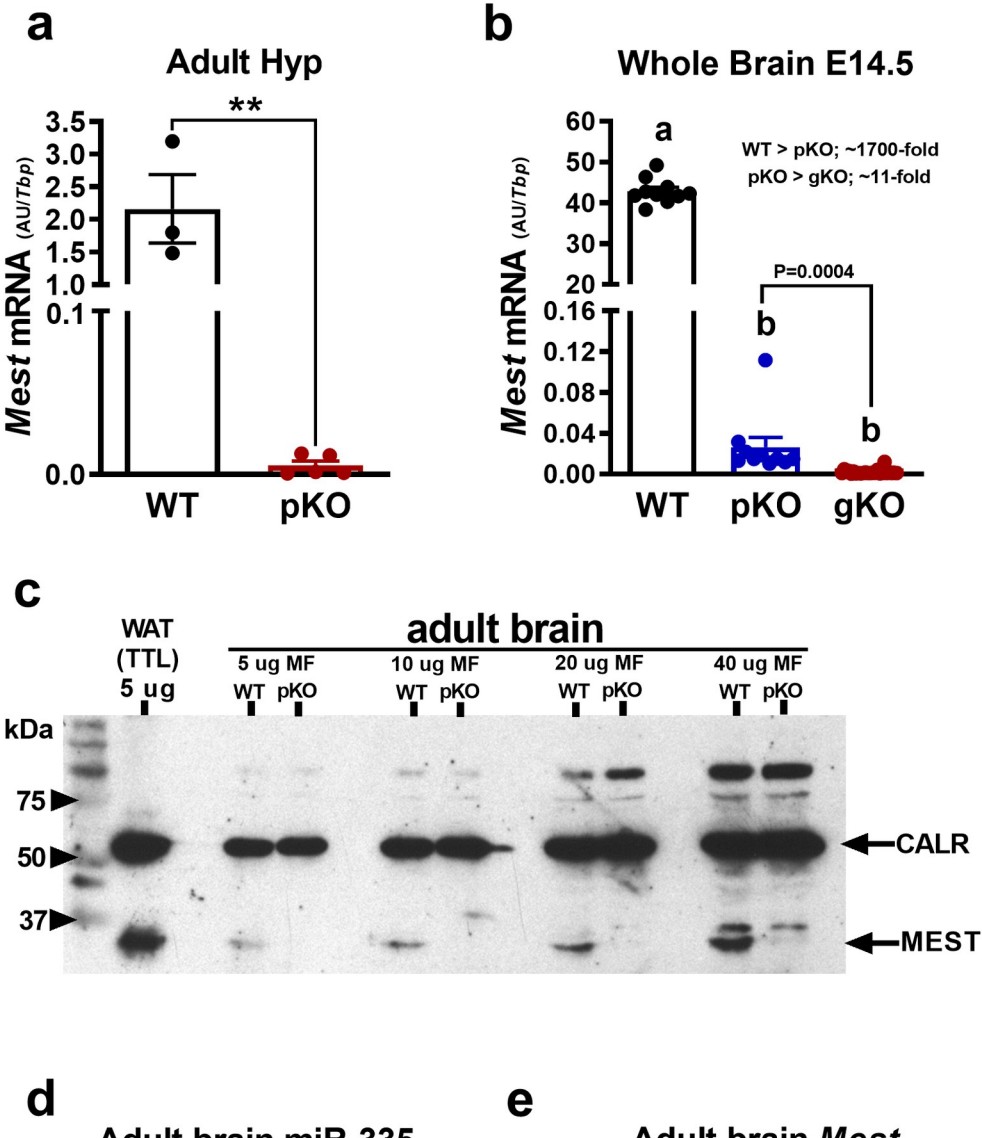

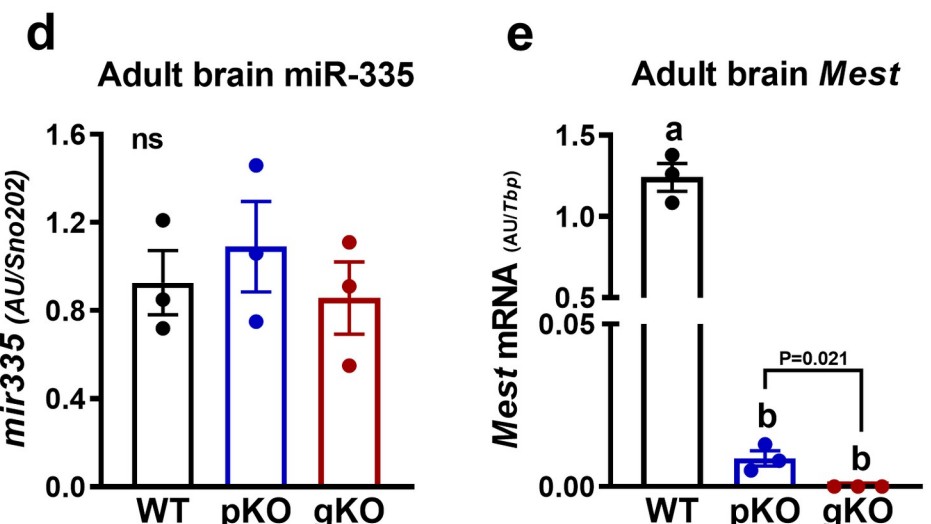

**Fig 5. *Mest* and *miR-335* in brain tissue.** Adult hypothalamic (a) and embryonic day 14.5 (E14.5) whole brain (b) showed the expected abrogation of *Mest* mRNA in *Mest*pKO and *Mest*gKO mice. Western blot analyses (c) shows loss of MEST in enriched membrane fractions (MF) of adult brain with white adipose tissue (WAT) total tissue lysate (TTL)

used as a control. Analyses of miR-335 (d) and corresponding *Mest* mRNA (e) expression in adult brain. Data was analyzed using unpaired t-test or multiple one-way ANOVA. Data with 2 asterisks (a) indicates P ≤ 0.01 using an unpaired t-test. Data sets analyzed via ANOVA (b, d and e) that do not share the same letter are significantly different from each other.

While digging behavior by dams is typically performed towards an intruder as an aggressive display, it is not clear why this phenotype was presented in the virgin female mice in our study. It is possible that this is a strain (e.g. C57BL/6J) dependent phenotype. Digging/burying of pups was observed more frequently in virgin WT females compared to virgin *Mest*^pKO and *Mest*^gKO females (Fig 3C) and may be due to differences in their neophobic response towards the pups [18], whereas digging behavior in the dams (Fig 4E) could be due to stress and anxiety caused by being separated from their pups.

In stark contrast to the *Mest* mutant mice developed by Surani's group [10], which showed poor reproductive fitness and abnormal maternal behavior, our model shows normal maternal instinct in virgin and primiparous *Mest*^pKO and *Mest*^gKO female mice compared to WT controls. In our model *(Mest*^tm1.2Rkz), virgins and first-time dams of both genotypes (*Mest*^pKO and *Mest*^gKO) displayed normal maternal behavior that is reflected in the breeding performance parameters previously reported for *Mest*^pKO [4], and in *Mest*^gKO as indicated in the Mouse Breeding and Study Design section of the Materials and Methods which showed robust litter sizes (~10 pups/litter) and a normal distribution of female and male pups in *Mest*^gKO X *Mest*^gKO breeding regimens. One caveat to our study is that maternal behavioral testing was first performed using virgin females that were subsequently mated one week after testing, and then retested as primiparous dams. While the primiparous dams may have retained some 'memory' of their 5 min exposure to pups while as virgins which could influence their maternal response at this later time, all genotypes were comparably tested and no significant differences were observed between genotypes.

A possible limitation in our study is that the time period used to identify the latency, or time to retrieve the first pup by virgin females, was shorter (300s) compared to the 900s used in studies by Lefebvre et al [10]. We selected 300s for pup retrieval based on observations from our initial virgin female cohort tested for 600s. With the exception of one WT and one *Mest*^pKO virgin female, all mice showing retrieval behavior completed pup retrieval within the first 300s of the test. Thus it was deemed that 300s was sufficient to detect pup retrieval in our study. In addition, it is important to note that both *Mest*^pKO and *Mest*^gKO virgin females in our study trended to retrieve pups more frequently during the 300s time period than WT females, whereas none of the paternal *Mest* knockout mice (*Mest*^tm1Masu) generated by Surani's group retrieved pups during a 900s period [10].

At this point, we can only speculate on the basis for the phenotypic differences in maternal behaviors between these two models (*Mest*^tm1Masu vs *Mest*^tm1.2Rkz). First, two different gene targeting strategies were employed in generating these mice. For the *Mest*^tm1Masu model, a bi-functional LacZ/neo fusion protein was inserted into the *Mest* locus leading to disruption of exons 3 to 8 and proximal exon 9 spanning 4.7 kb in length [10]. *Mest*^tm1.2Rkz was generated using Cre/loxP technology to insert loxP sites flanking exon 3 and subsequent Cre-recombinase deletion of the floxed exon 3 resulted in a downstream frameshift mutation and the generation of a stop codon in *Mest* exon 4 [4]. Second, these two strategies may differ on their impact on the *Mest* gene promoter and regulatory elements in genomic proximity. Introduction of the IRES-βgeo cassette disrupted a larger sequence of the *Mest* gene (exons 3 to 9) in the *Mest*^tm1Masu model whereas we focused on deletion of only exon 3 of the *Mest* gene leading to a truncated MEST protein. It has been previously reported in a gene deletion model for

paternally expressed gene 3 (*Peg3*), another imprinted gene, that the insertion of large expression cassettes could have potential side effects on the locus itself and/or function of the adjacent imprinted genes [19]. A recent study using two mouse models, one expressing bacterial LacZ and the other without a LacZ reporter, to study the inactivation of the voltage gated potassium channel subunit *Kcna6* showed that the LacZ reporter may be neurotoxic to sensory neurons [20]. Therefore, it is plausible that the LacZ reporter in the *Mest*[tm1Masu] strain could be eliciting some of the maternal behavioral changes in these mice [10].

The *Mest* locus also harbors miR-335, a microRNA within intronic sequence between exons 2 and 3 [21]. Although miR-335 genomic locus was left intact in *Mest*[tm1Masu] mice based on the gene targeting strategy, miR-335 expression in the skeletal muscle was significantly lower in *Mest*[tm1Masu] with a paternal allelic inactivation compared to WT mice [22]. On the other hand, miR-335 expression in *Mest*[tm1.2Rkz] mice was comparable between WT and mice with a paternal inactivation of *Mest* in adipose tissue [4]. Consistent with the latter observation, in the present study we have also demonstrated that miR-335 expression is unaffected in the adult brain of mice with either paternal or global inactivation of *Mest* (Fig 5D). The difference in miR-335 gene expression between these two models in relation to *Mest* expression suggests a more broad disruption of the *Mest* locus with the *Mest*[tm1Masu] compared with *Mest*[tm1.2Rkz] mice. Third, another possible reason could be the difference in the genetic backgrounds between these two mutant mice. *Mest*[tm1.2Rkz] is fully congenic on the C57BL/6J genetic background [4] whereas *Mest*[tm1Masu] showed consistent abnormal maternal behavior regardless of the genetic background (129/SvEV, C57BL/6, Balb/c) or (C57BL/6 x CBA/Ca)F1 [10]. A recent publication reported the re-acquisition of normal reproductive behavior in a colony of *Mest*[tm1Masu] mutant model after backcrossing the original mutant five times to 129/SvJ background due to breeding difficulties [23]. One possible explanation for the normal maternal behavior observed in this study is the use of a different 129 substrain (129/SvJ) for breeding compared to the original background (129/SvEV). These two substrains (129/SvJ vs 129/SvEV) diverged in the 1960s and are skin graft incompatible [24]. In this recent study by Ineson et al [23], the original gene targeting construct appeared to be fully functional based on the LacZ expression in those mice with a disruption of paternal *Mest*, and no expression in WT mice, which suggests that the phenotypic reversion is due to other factors besides the *Mest* mutation itself.

In summary, based on the normal social and maternal behavior we observed in our mutant mouse, it is possible that *Mest*, although highly expressed in the brain, plays a limited role in determining social, object recognition memory and maternal behavior in mice.

## Supporting information

**S1 Table. Tail suspension test.**
(DOCX)

**S1 File. Raw data and analyses; behavioral studies.**
(XLSX)

**S1 Raw images. Western blot for MEST; raw image.**
(PDF)

## Acknowledgments

We thank Crystal L. Bilodeau for assistance in mouse colony management.

## Author Contributions

**Conceptualization:** Rea P. Anunciado-Koza, J. Patrizia Stohn, Arturo Hernandez, Robert A. Koza.

**Data curation:** J. Patrizia Stohn, Robert A. Koza.

**Funding acquisition:** Robert A. Koza.

**Investigation:** Rea P. Anunciado-Koza, J. Patrizia Stohn.

**Project administration:** Robert A. Koza.

**Supervision:** Robert A. Koza.

**Writing – original draft:** Rea P. Anunciado-Koza, J. Patrizia Stohn.

**Writing – review & editing:** Arturo Hernandez, Robert A. Koza.

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
