## [Decision Letter · Decision Letter 0]

18 May 2022

PONE-D-22-11179Social and maternal behavior in mesoderm specific transcript (Mest)-deficient micePLOS ONE

Dear Dr. Koza,

Thank you for submitting your manuscript to PLOS ONE. After careful consideration, we feel that it has merit but does not fully meet PLOS ONE’s publication criteria as it currently stands. Therefore, we invite you to submit a revised version of the manuscript that addresses the points raised during the review process.

Both reviewers agree that the work was performed well and clearly reported.  The differences in behavior found between the two MEST KO  alleles is an important finding, that both reviewers felt was handled well in the discussion.   Reviewer #1 felt it important to include data that verifies the absence of MEST in the new allele. This work was previously reported, although showing the specific loss of the protein within the brain would strengthen the article, I do not believe this is required.   Nevertheless, this reviewer does bring up doubts regarding the effect of the new KO allele that could be exists with other reviewers.  Further highlighting the results of this KO on MEST expression with a more complete description, or the addition of supplementary material would be helpful.  

 Reviewer #2 had more detailed comments regarding the descriptions and discussion of the behavioral tests and reporting of the results.  I find these comments to extremely helpful for improving the manuscript and preparing it for publication.   Importantly, they do not appear to require any additional experimentation, and do not impact the validity of the results.  For resubmission, please carefully consider this reviewers comments on the discussion and description of the different behaviors used, and the difficulties with the time scales.   Also, please be sure to report the outcomes of the statistical analysis within the RESULTS section.  

We look forward to receiving your revised manuscript.

Kind regards,

Gregg Roman, PhD

Academic Editor

PLOS ONE

Journal Requirements:

Reviewers' comments:

Reviewer's Responses to Questions

**Comments to the Author**

1. Is the manuscript technically sound, and do the data support the conclusions?

Reviewer #1: Partly

Reviewer #2: Partly

2. Has the statistical analysis been performed appropriately and rigorously? 

Reviewer #1: I Don't Know

Reviewer #2: No

3. Have the authors made all data underlying the findings in their manuscript fully available?

Reviewer #1: Yes

Reviewer #2: No

4. Is the manuscript presented in an intelligible fashion and written in standard English?

Reviewer #1: Yes

Reviewer #2: Yes

5. Review Comments to the Author

Reviewer #1: The authors reported normal social, maternal and cognitive functions in a new Mestp/gKO line. It is interesting and informative, as they show stark contrast of decreased survival and severe behavioral abnormalities in a previously-established MestKO. Because of the importance of Peg1 and relevant issues in scientific field, I endorse publication of this article in Plos One.

Two things to be added before publication is the confirmation of the absence of Mest/Peg1 gene or gene product from the KO; and more description about survival and general health of the present strain of MestKO. Because the differences in their health condition are very large in these two lines, the reader may doubt the proper deletion of the gene. At least more thorough discussion, and a possible limitation of the present study should be provided to the text.

Reviewer #2: In this manuscript the authors describe a series of experiments to test various aspects of behaviour in a new Peg1/Mest KO mouse model (Mesttm1.2Rkz). The rationale for doing this experiment is that in breeding the Mesttm1.2Rkz line, the authors found no deviation in expected ratios of the different genotypes, which is in contrast to studies of a previously generated line (Mesttm1Masu) which was due to poor maternal care by females carry a paternal deletion. Here, the authors test a number of different behaviours, including maternal care, in the new Mesttm1.2Rkz line, examining both paternal and full (i.e. homzygote) knockouts. The authors find no difference between the groups in measures of anxiety / memory / sociability or maternal behaviour, despite clear loss of Mest mRNA in the hypothalamus and embryonic brain.

Generally, the behavioural analysis is robust and well-carried out. There are a few issues I think need to be addressed however. Firstly the number of behavioural test performed by each animal is not ideal, the key problem being the timescale within which all these behaviours were peformed - six separate tests (some last >1 day) in ~20 days is not ideal, especially as some (EPM, tail suspension) are quite aversive. I do not expect the authors to rectify this, as I doubt it will affect the outcome, but I think it needs to be acknowledged in some way in the Discussion. Another thing that is not clear from the methods is the relation of this batch of tests to the maternal behaviour testing in virgins and subsequently primiparous mums. I assume the females tested in the battery of tests were the same ones used in maternal behaviour testing? I also think it needs to be made clearer that the virgin females were then mated (how long afterwards?) and re-tested as primiparous mums. This also needs to be briefly discussed in the Discussion, as mice (including virgins) will improve maternal behaviour with subsequent tests, and so the primiparous mums have effectively had a practice run whilst virgins. This is a confound, however I doubt it will have consequences for the overall findings.

The authors refer to "cognitive function' throughout. In fact this actually refers to one test, the novel object test. As there is not a more comprehensive wide-ranging set of cognitive tests, I would suggest simply referring to this domain of behaviour as "object recognition memory" (or, at a stretch, "memory").

There seems to be no statistics presented in the Results and / or figure legends. If ANOVA has been used (as indicated in the Methods) then I would expect to see an F-statistic and degrees of freedom, not simply an asterisk on a graph. This makes it very difficult to judge what test has been used to test each set of data. For instance both the "nest quality" and the latency data are not normally distributed (the latency has an upper bounding of 300s, which limits the Virgin data in particular) and so should not be analysed by ANOVA - but it is difficult to tell HOW they have been analysed. Again, looking at the data (which are nicely presented in the figures) I doubt this will effect the outcome, but it is important it is correct.

The central focus of this work is on the maternal behaviour and, although it is clear there are no genotype differences in this behaviour in dams, I think some of the other inferences need clarity. The test itself is only 5-minutes (300s) long. 5 mins is a very short time period for retrieval, particularly for the virgin females as this window would not have given them adequate time to display parenting – hence the large amounts of NR in Figure 3. Is there any reason the authors didn’t allow longer, say 15 minutes like the original Surani paper. Either way, this limitation needs to be acknowledged in the discussion. Similarly, on line 321 the authors refer to digging behaviour in this test as ‘aggressive behaviour’ – is there evidence to support this being classified this way and not just an exploratory behaviour? Relatedly, are there any behaviour measures of what the mice were doing during the 5 minute retrieval for comparisons i.e. did the WT’s spend less time interacting with the pups or more time exploring or in the nest alone? Adding these could reveal differences or strengthen the argument that there are none. Finally, the assertion that maternal instinct appears ‘enhanced’ is a bit of a stretch considering that there are no significant differences between the groups and the ‘mean differences’ will be largely influenced by the one failed WT dam retrieval within the time limit. Since all the animals began to retrieve their pups within a short time window (< 5 mins) I don’t think there is any good evidence of an ‘enhancement in maternal instinct’ here, and this should be removed from the text.

The Discussion is focused on explaining why these findings in the Mesttm1.2Rkz line deviate so dramatically from the those seen in the Mesttm1Masu line. Here, the authors do an excellent job in outlining many of the possible explanations. One thing that is touched upon, but not considered in detail, is the effect of the presence of LacZ in the Mesttm1Masu - LacZ has been shown to have effects on neuronal development and function previously (e.g. https://doi.org/10.1523/JNEUROSCI.0187-21.2021). A key issue the authors highlight could be the lack of an effect of the deletion in the Mesttm1.2Rkz on the intronic miR-335. I wonder why the authors did not look at the expression of miR-335 in the brain? This would easy given the available RNA (Fig. 5).

Minor issues:

Figure 2 – confusing when the colour scheme for female (red) and male (blue) is used to now separate stranger (red) and empty (blue). Shades of red for female only graphs and shades of blue for male only graphs would keep the colour scheme consistent and still distinguish stranger vs empty.

Figure 3/4 - would be good to see latency clarified (I’m assuming it is latency to retrieve the first pup to the nest? But its not clearly stated)

6. PLOS authors have the option to publish the peer review history of their article (what does this mean?). If published, this will include your full peer review and any attached files.

Reviewer #1: No

Reviewer #2: **Yes: **Anthony R. Isles

---

## [Author Response · Author response to Decision Letter 0]

1 Jul 2022

Reviewer #1: 

Two things to be added before publication is the confirmation of the absence of Mest/Peg1 gene or gene product from the KO; and more description about survival and general health of the present strain of MestKO. Because the differences in their health condition are very large in these two lines, the reader may doubt the proper deletion of the gene. At least more thorough discussion, and a possible limitation of the present study should be provided to the text.

To address these concerns, we analyzed MEST protein in membrane fractions (enriched for endoplasmic reticulum) of brain tissue from adult WT and paternal knockout (pKO) mice and show an absence of MEST protein in the pKO mice. These data are consistent with our previously published results in PloS One in 2017 (pmid: ) showing complete loss of Mest mRNA and MEST protein in adipose tissue of pKO mice. We have previously demonstrated that MEST protein predominantly co-localizes with the endoplasmic reticulum in tissue and cells of mice (Nikonova et al 2008, Prudovsky et al 2018). The western blot data for the brain will be included as part of Figure 5 (Fig 3c) in the revised manuscript. 

Additional discussion regarding possible reasons for the differences in the health and behavior between the Mesttm1Masu and our Mesttm1.2Rkz strain of Mest knockout mice are included in the manuscript. 

Reviewer #2: 

Generally, the behavioural analysis is robust and well-carried out. There are a few issues I think need to be addressed however. Firstly the number of behavioural test performed by each animal is not ideal, the key problem being the timescale within which all these behaviours were peformed - six separate tests (some last >1 day) in ~20 days is not ideal, especially as some (EPM, tail suspension) are quite aversive. I do not expect the authors to rectify this, as I doubt it will affect the outcome, but I think it needs to be acknowledged in some way in the Discussion.

While batteries of tests are often used to investigate complex and multivariate neurological diseases/behaviors that cannot simply be described by a single test, we are aware that the individual tests might have an effect on each other as well as the repeated handling of the animals (Voikar et al 2004; pmid 14960013). Therefore, we attempted to choose the order in which the tests were conducted to reduce the potential that they effect on each other. We agree with the reviewer’s suggestion and have acknowledged that the comprehensive testing of mice within a 20-24 day period may not be ideal within the discussion.

Another thing that is not clear from the methods is the relation of this batch of tests to the maternal behaviour testing in virgins and subsequently primiparous mums. I assume the females tested in the battery of tests were the same ones used in maternal behaviour testing? I also think it needs to be made clearer that the virgin females were then mated (how long afterwards?) and re-tested as primiparous mums. This also needs to be briefly discussed in the Discussion, as mice (including virgins) will improve maternal behaviour with subsequent tests, and so the primiparous mums have effectively had a practice run whilst virgins. This is a confound, however I doubt it will have consequences for the overall findings.

The methods section for Behavioral Testing has been edited to clearly indicate that maternal behavior testing in virgin females was performed 2 weeks after completion of behavioral and social testing and virgin females were then mated 1 week later. Tests for maternal behavior in virgin females are typically performed on consecutive days (e.g. 4 days, Stolzenberg & Rissman, 2011, PMID: 21276101). However, we found no data suggesting that these mice retain this experience and, because of this, show increased maternal behavior. Furthermore, we only exposed virgin females to the pups for a short time (5 min), which is not likely to have a long-term effect on their display of maternal behavior. We have also addressed this concern in the discussion as per suggestion by the reviewer.

The authors refer to "cognitive function' throughout. In fact this actually refers to one test, the novel object test. As there is not a more comprehensive wide-ranging set of cognitive tests, I would suggest simply referring to this domain of behaviour as "object recognition memory" (or, at a stretch, "memory").

This is a great suggestion. We have changed the description of cognitive function’ to ‘object recognition memory’ throughout the manuscript.

There seems to be no statistics presented in the Results and / or figure legends. If ANOVA has been used (as indicated in the Methods) then I would expect to see an F-statistic and degrees of freedom, not simply an asterisk on a graph. This makes it very difficult to judge what test has been used to test each set of data. For instance both the "nest quality" and the latency data are not normally distributed (the latency has an upper

bounding of 300s, which limits the Virgin data in particular) and so should not be analysed by ANOVA - but it is difficult to tell HOW they have been analysed. Again, looking at the data (which are nicely presented in the figures) I doubt this will effect the outcome, but it is important it is correct.

We have now added an ANOVA F statistic and degrees of freedom for relevant analyses within the manuscript. We agree with the reviewer that the 300s (5 min) period for latency in pup retrieval, which was not met by a significant number of virgin females in this study, makes it difficult to appropriately analyze via ANOVA. To address this, we now just present the data from mice that showed at least partial retrieval within the 300s time-period in Fig 3b. Analyses of these data via ANOVA show no significant differences between strains. 

The central focus of this work is on the maternal behaviour and, although it is clear there are no genotype differences in this behaviour in dams, I think some of the other inferences need clarity. The test itself is only 5-minutes (300s) long. 5 mins is a very short time period for retrieval, particularly for the virgin females as this window would not have given them adequate time to display parenting – hence the large amounts of NR in Figure 3. Is there any reason the authors didn’t allow longer, say 15 minutes like the original Surani paper. Either way, this limitation needs to be acknowledged in the discussion.

The studies by Lefebvre et al (PMID: 9302270) focused on three aspects of maternal behavior: latency to detect and sniff one of the pups, time required to retrieve one of the pups, latency to initiate the construction of a nest with wood chips. We tested separately for the nest building using nestlets as nesting material. Initially we tested the first cohort of virgins for pup retrieval for 600s. However, with the exception of one WT and one pKO female, the latency for retrieval and the time until the first pup was in the nest was less than 300s. Prolonging the test by 300s did not appear to have an effect on subsequent retrieval. Since in our experience WT mothers retrieve their pups almost at the start of the test and are typically done within less than a minute with retrieving all pups to the nest, we limited the tests to 5 mins. While Surani’s group focused on the latency to detect and sniff one of the pups, we noticed that this was an almost immediate behavior in all of the virgins and dams, however at the beginning of the test it was also very brief and often in passing as the female seemed to quickly assess the situation. Therefore we didn’t measure the latency for this. We agree with the reviewer that, in retrospect, allowing virgin females a 15-minute period may have been a better approach, however, it probably would not have changed the outcome or interpretation of our current analyses. We will acknowledge this limitation in our study as part of the discussion.

Similarly, on line 321 the authors refer to digging behaviour in this test as ‘aggressive behaviour’ – is there evidence to support this being classified this way and not just an exploratory behaviour? Relatedly, are there any behaviour measures of what the mice

were doing during the 5 minute retrieval for comparisons i.e. did the WT’s spend less time interacting with the pups or more time exploring or in the nest alone? Adding these could reveal differences or strengthen the argument that there are none.

The digging around and/or burying of pups is a behavior we have not normally seen in our mice and we have not found this particular behavior described elsewhere in the literature. It could be due to conditions during the study or is possibly strain (e.g. C57BL/6J) dependent. When referring to maternal aggression it is typically the behavior displayed by the dam towards an intruder. In our experience with the resident-intruder paradigm test, we have observed similar digging behavior in the resident and classified it as threat or aggressive behavior. Martin-Sanchez et al, 2015 (PMID: 26257621) investigated the induction of maternal care in virgin female CD1 mice and indicated risk-assessment as part of their behavioral measures. Part of that were studies of a neophobic response towards the pups similar to that expressed by virgin rats (Fleming and Luebke, 1981 PMID: 7323193) during the sensitization period. Therefore, we suggest to refer to this behavior as neophobic burying brought on by novelty-induced anxiety. Unfortunately, we have not detected additional obvious behavioral measures during this 5 min period except to note that the WT virgin females tended to show increased digging/burying of pups compared to strains with paternal or global inactivation of Mest. 

Finally, the assertion that maternal instinct appears ‘enhanced’ is a bit of a stretch

considering that there are no significant differences between the groups and the ‘mean differences’ will be largely influenced by the one failed WT dam retrieval within the time limit. Since all the animals began to retrieve their pups within a short time window (< 5 mins) I don’t think there is any good evidence of an enhancement in maternal instinct’ here, and this should be removed from the text.

We agree with the reviewer and have now removed this wording from the text.

One thing that is touched upon, but not considered in detail, is the effect of the presence

of LacZ in the Mesttm1Masu - LacZ has been shown to have effects on neuronal development and function previously (e.g. https://doi.org/10.1523/JNEUROSCI.0187-21.2021).

We thank the reviewer for the reference regarding LacZ effects on neuronal development. We have cited this study in our discussion.

A key issue the authors highlight could be the lack of an effect of the deletion in the Mesttm1.2Rkz on the intronic miR-335. I wonder why the authors did not look at the expression of miR-335 in the brain? This would easy given the available RNA (Fig. 5).

We agree with the reviewer and have now analyzed miR-335 in the brain of adult mice. Data are presented in Fig 5. As observed in prior studies of adipose tissue, inactivation of Mest in our mice had no effect on miR-335 expression in the brain. These studies were complicated by the need to re-isolate RNA from tissue using methods to optimize purification of microRNA’s, however, it was deemed an important contribution to the manuscript.

Minor issues:

Figure 2 – confusing when the colour scheme for female (red) and male (blue) is used to now separate stranger (red) and empty (blue). Shades of red for female only graphs and shades of blue for male only graphs would keep the colour scheme consistent and still distinguish stranger vs empty.

We agree with the reviewer and the color scheme of Figure 2 was modified to keep all females red and males blue.

Figure 3/4 - would be good to see latency clarified (I’m assuming it is latency to retrieve the first pup to the nest? But its not clearly stated).

We will add a clear statement indicating that this is indeed the latency to retrieve the first pup to the next. Thank you for your input here.

---

## [Editor Report · Decision Letter 1]

11 Jul 2022

Social and maternal behavior in mesoderm specific transcript (Mest)-deficient mice

PONE-D-22-11179R1

Dear Dr. Koza,

We’re pleased to inform you that your manuscript has been judged scientifically suitable for publication and will be formally accepted for publication once it meets all outstanding technical requirements.

Kind regards,

Gregg Roman, PhD

Academic Editor

PLOS ONE
---

## [Editor Report · Acceptance letter]

13 Jul 2022

PONE-D-22-11179R1 

Social and maternal behavior in mesoderm specific transcript (*Mest*)-deficient mice 

Dear Dr. Koza:

I'm pleased to inform you that your manuscript has been deemed suitable for publication in PLOS ONE. Congratulations! Your manuscript is now with our production department. 

Kind regards, 

on behalf of

Dr Gregg Roman 

Academic Editor

PLOS ONE